# Optical Identification of Diabetic Retinopathy Using Hyperspectral Imaging

**DOI:** 10.3390/jpm13060939

**Published:** 2023-06-01

**Authors:** Ching-Yu Wang, Arvind Mukundan, Yu-Sin Liu, Yu-Ming Tsao, Fen-Chi Lin, Wen-Shuang Fan, Hsiang-Chen Wang

**Affiliations:** 1Department of Ophthalmology, Dalin Tzu Chi Hospital, Buddhist Tzu Chi Medical Foundation, Chiayi 62247, Taiwan; s19001052@gmail.com (C.-Y.W.); wsfan@tzuchi.com.tw (W.-S.F.); 2Department of Mechanical Engineering, National Chung Cheng University, Chiayi 62102, Taiwan; d09420003@ccu.edu.tw (A.M.); dan102030440@gmail.com (Y.-S.L.); d09420002@ccu.edu.tw (Y.-M.T.); 3Department of Ophthalmology, Kaohsiung Armed Forces General Hospital, Kaohsiung 80284, Taiwan; 4Director of Technology Development, Hitspectra Intelligent Technology Co., Ltd., Kaohsiung 80661, Taiwan

**Keywords:** hyperspectral imaging, fundus imaging, principal component analysis

## Abstract

The severity of diabetic retinopathy (DR) is directly correlated to changes in both the oxygen utilization rate of retinal tissue as well as the blood oxygen saturation of both arteries and veins. Therefore, the current stage of DR in a patient can be identified by analyzing the oxygen content in blood vessels through fundus images. This enables medical professionals to make accurate and prompt judgments regarding the patient’s condition. However, in order to use this method to implement supplementary medical treatment, blood vessels under fundus images need to be determined first, and arteries and veins then need to be differentiated from one another. Therefore, the entire study was split into three sections. After first removing the background from the fundus images using image processing, the blood vessels in the images were then separated from the background. Second, the method of hyperspectral imaging (HSI) was utilized in order to construct the spectral data. The HSI algorithm was utilized in order to perform analysis and simulations on the overall reflection spectrum of the retinal image. Thirdly, principal component analysis (PCA) was performed in order to both simplify the data and acquire the major principal components score plot for retinopathy in arteries and veins at all stages. In the final step, arteries and veins in the original fundus images were separated using the principal components score plots for each stage. As retinopathy progresses, the difference in reflectance between the arteries and veins gradually decreases. This results in a more difficult differentiation of PCA results in later stages, along with decreased precision and sensitivity. As a consequence of this, the precision and sensitivity of the HSI method in DR patients who are in the normal stage and those who are in the proliferative DR (PDR) stage are the highest and lowest, respectively. On the other hand, the indicator values are comparable between the background DR (BDR) and pre-proliferative DR (PPDR) stages due to the fact that both stages exhibit comparable clinical-pathological severity characteristics. The results indicate that the sensitivity values of arteries are 82.4%, 77.5%, 78.1%, and 72.9% in the normal, BDR, PPDR, and PDR, while for veins, these values are 88.5%, 85.4%, 81.4%, and 75.1% in the normal, BDR, PPDR, and PDR, respectively.

## 1. Introduction

All the anatomical structures need to be examined and understood for the prognosis of all the relevant diseases in the retina [1,2]. Specifically, classifying the retinal vessels into veins and arteries will act as an important biomarker for various eyesight-related disorders [3]. Some cardiovascular diseases, diabetes, and various other diseases will cause variations in the natural diameters of these veins and arteries in the early stages of these diseases [4,5]. However, if these diseases are diagnosed in their early stages, the survival rate of the patients can be drastically increased [6]. During the early stages of the classification of arteries and veins, traditional image processing techniques were used along with artificial neural networks (ANN) [7,8,9,10]. After the introduction of deep learning, convolution neural networks (CNN) greatly increased the accuracy of the models [11,12,13]. However, in recent years, fully convolution neural networks (FCNN) have begun being employed along with various architectures such as U-net, Res-net, and Fusion-net for segmentation of arteries and veins in fundus imaging [14,15,16,17].

However, one of the optical imaging applications that have not been employed in fundus imaging is hyperspectral imaging (HSI). In HSI, the data from the image are not only processed in the three visible colors but in the whole electromagnetic spectrum. HSI has been previously employed in agriculture [18], astronomy [19], military [20], biosensors [21,22,23], air pollution detection [24,25], remote sensing [26], dental imaging [27], environment monitoring [28], satellite photography [29], cancer detection [30,31,32], forestry monitoring [33], security [34], food security [35], natural resources surveying [36], vegetation observation [37], and geological mapping [38]. DR (DR) is one of the major complications of microvascular injury [39]. In its initial stages, DR is asymptomatic. However, as in its latter stages, it can cause complete loss of vision. Currently, almost 28 million people around the world have lost their vision due to DR and by the year 2030, 51 million people out of the 155 million people affected by DR will lose their vision [40]. One of the major reasons for the loss of vision is DR in the working ages between 20 and 60 [41]. Therefore, early diagnosis and medication of DR are necessary to avoid vision loss. DR can be usually classified into three stages: BDR, PPDR and PDR.

This study proposes a novel technique of using hyperspectral imaging to collect the database of spectra combined with a Gabor filter for vessel classification; finally, principal component analysis is carried out to train the model to distinguish between the different stages of DR. In this study, the images are classified into four different categories based on the recommendations of the International Council of Ophthalmology (ICO); normal, background DR (BDR), pre-proliferative DR (PPDR), and proliferative DR (PDR) [42,43].

## 2. Materials and Methods

In this study, a total of 91 patients with 28 BDR, 28 PPDR, 35 PDR and 35 images with normal vision (without associated systemic or ophthalmological pathology) were utilized. Out of the 126 images, 58 images are from male patients (15 in normal, 13 in BDR, 10 in PPDR and 20 in PDR), while 68 images are from female patients (20 in normal, 15 in BDR, 18 in PPDR and 15 in PDR). The major exclusion criterion was images being blurred or having noise caused by the light source, leading these to be omitted from the study. The second exclusion criterion is that the patients who had cataract-related or any other forms of surgery are excluded. The major inclusion criteria for this study is that patients within the age range of 20 and 60 years treated with DR are accepted. In this study, all the patients had type II diabetes, which is relatively more common than type I [44]. In this study, the longest time a patient had been diagnosed with diabetes was 25 years and the shortest was 1 year, while the median was 7 years, thereby encompassing different years since the diagnosis of diabetes. The data from the test set constituted thirty percent of the total data in the study. The images of the fundus were captured with the help of a Zeiss fundus camera (Carl Zeiss AG, Jena, Germany). The principle of retinal fundus camera is shown in Appendix A [45]. Imaging, lighting, and observational systems are all contained within the fundus camera. The design of the imaging system consisted of three components: the eye objective lens, the imaging objective lens, and the negative film. These components were designed in a manner that was distinct from the imaging mode of conventional cameras. This was carried out to prevent the influence of other factors, such as reflected light on the cornea. The lighting system comprised two different light sources, the first of which was used for lighting the fundus while the camera was being focused, and the second of which was the flash, which was used to increase the brightness of the fundus lighting when the picture was being taken. Other methods used to observe fundus images include fundus fluorescein angiography, and optical coherence tomography as shown in Appendix A [46].

The experimental flow diagram for this study can be found in Figure 1. First, the blood vessels in the fundus images were captured, and then the background was removed from the image so that the blood vessels could be seen more clearly. After that, spectral data were constructed with the help of the HSI technique. At the same time, PCA was utilized to obtain principal components score plots for retinopathy in arteries and veins at all stages. In the final step, arteries and veins in the original fundus images were separated using the principal components score plots for each stage.

In this particular study, Matlab (The MathWorks, Inc., Natick, MA, USA) was utilized for the pre-processing of images, while other parts were performed in Python 3.11, (the Python Software Foundation, Wilmington, DE, USA), which was utilized for PCA training. In order to obtain only the required data, it is necessary to process the image first in order to obtain the distribution diagram of all blood vessels in the retina so as to obtain the spectral data of the arteries and veins in the fundus image. The entirety of the experimental procedure can essentially be broken down into three distinct stages. The first component is the algorithm for processing images of the retina. After the blood vessels in the fundus image have been preprocessed, the blood vessels of the fundus will be segmented by removing the background. Following this, HSI technology will be used to obtain the spectral data in order to analyze and simulate the overall reflection spectrum of the retinal image. A linear transformation of the reflection spectrum that was obtained in the second part is then used in the final step of the PCA process, which is used to reduce the original feature dimension while retaining as much as possible. After obtaining the arterial and venous principal component score maps of each stage of retinopathy using the differences in the original features, the arterial and veinous structures in the original fundus image were differentiated using the principal component score maps of each stage.

Figure 2 demonstrates that the algorithm can be broken down into these four distinct steps. In the first step of the process, pre-processing was used to emphasize the blood vessels, small noise was binarized and removed (as it is in RGB images), and the green component of the image was extracted for pre-processing. This was carried out because the green component of the image has obvious contrasting characteristics in comparison to the other two different components. In the second step of the process, a Gabor Filter was applied to the image in order to improve the data on blood vessels so that they could be seen more clearly. The schematic diagram of a two-dimensional Gabor Filter was shown in Appendix A [47]. Gabor filter is a linear filter commonly used for edge detection. A two-dimensional Gabor filter is a Gaussian function modulated by sinusoidal plane waves. Given that its representation in the spatial domain and frequency domain is similar to human biological vision specificity, it can intuitively describe the structural information, such as the spatial position of images, directional selectivity, and spatial frequency.

The mathematical equation of the Gabor filter can be expressed as Equations (1) and (2).
(1)$$g\left(x,y\right)=\frac{1}{2\pi \sigma_{x}\sigma_{y}}exp\left[-\frac{1}{2}\left(\frac{x^{\prime 2}}{\sigma_{x}^{2}}+\frac{y^{\prime 2}}{\sigma_{y}^{2}}\right)\right]\cos\left(2\pi f_{0}x^{\prime }\right)$$
(2)$$g\left(x,y\right)=\frac{1}{2\pi \sigma_{x}\sigma_{y}}exp\left[-\frac{1}{2}\left(\frac{x^{\prime 2}}{\sigma_{x}^{2}}+\frac{y^{\prime 2}}{\sigma_{y}^{2}}\right)\right]\sin\left(2\pi f_{0}x^{\prime }\right)$$

In particular, $g\left(x,y\right)$ is the response value of the image that is processed by the Gabor filter; $f_{0}$ is the frequency of the curve; and $\sigma_{x}$ and $\sigma_{y}$ are the standard deviations in the x and y directions, respectively.

For the edge extraction of fundus images, a series of filter responses can be obtained from the results of all angles in the range of $\left[\frac{-\pi }{2},\frac{\pi }{2}\right]$ through the Gabor filter. The results of the coordinate transformation formula at different angles of rotation are shown in Equations (3) and (4).
(3)$$x^{\prime }\;=\;x\cos\theta \;+\;y\sin\theta$$
(4)$$y^{\prime }\;=\;-x\sin\theta \;+\;y\cos\theta$$

In this equation, ($x^{\prime },y^{\prime }$) are the corresponding coordinates after the rotation of θ values. Various filter responses $G^{\theta }\left(x,y\right)$ can be obtained using Gabor filters $g\left(x,y\right)$ with different θ values in fundus images. Then, the absolute values of the real and imaginary components of each angle are added, and the square root is taken, as shown in Equation (5).
(5)$$sqrt{(G_{\gamma }^{\theta }}^{2}+{G_{i}^{\theta }}^{2})\;=\;G^{\theta }$$

In the equation, $G_{\gamma }^{\theta }$ is the real component response, and $G_{i}^{\theta }$ is the imaginary component response. In order to obtain the vascular position more easily, only the maximum response $R\left(x,y\right)$ is retained in each pixel point $\left(x,y\right)$, as shown in Equation (6).
(6)$$R\left(x,y\right)\;=\;Max(G^{\theta }\left(x,y\right)),\;\theta \;=\;[-\frac{\pi }{2}:\frac{\pi }{180}:\frac{\pi }{2}]$$

In the third step of the process, an iterative binarization technique was utilized to locate image thresholds and differentiate vascular regions. In the end, some small white spots and noise were removed by filtering the image using the equations that were provided by the program. In this particular investigation, an iterative-based binarization method is utilized in order to locate the image’s threshold (Appendix A). Because the grayscale image ranges from 0 to 255, a threshold needs to be selected that not only isolates the area containing the blood vessels but also minimizes the influence of the background and any noise that may be present. If the grayscale distribution of the pixels in the target area is uniform, and the pixels in the background area are also uniformly distributed in another grayscale, then the corresponding gray value in the middle of these two peaks are chosen to be used as the threshold value, and the iterative method is to obtain this result by repeatedly applying the fixed sigma value. When the image to be processed consists of a target with a large difference in gray value and the rest of the background, then the initial threshold value is calculated by averaging the gray value of the entire image to get a starting point. Secondly, by dividing the image into two groups, the average gray value of each group can be calculated separately and these can then be averaged to obtain a new starting point. Finally, the previous steps are repeated in order to continuously generate a new threshold to obtain the difference between the thresholds that have come before. The final threshold is obtained when the difference is smaller than a parameter that has been set in advance. It is unavoidable that there will be some small white spots due to uneven brightness or poor quality in the image when setting the threshold value in the fourth step. Because of this, it will be divided into the same area as the blood vessels, which will have an effect on the identification that is ultimately performed. This will have an effect on the final result. This function can be used to set the connected objects with a number of pixels that is lower than the number of many pixels to delete, and the pixel connectivity is present when the function is first called.

After that, the hyperspectral image processing was carried out in three distinct stages, as can be seen in Figure 3. In the first step toward obtaining the relationship matrix between the fundus camera and the spectrometer, the data of the light source spectrum for the two instruments as well as the data of the reflection spectrum for the light source and 24 color-checker in the visible band (380–780 nm) were collected. This was carried out so that the relationship matrix could be obtained. In addition to the data, the standard 24-point color checker was used as the subject for measurement, which contributed to the increased accuracy of the transformation. Second, spectral analysis was carried out, and PCA was utilized in order to simplify the data. For the purpose of determining the eigenvalues, the eigenvectors of the first six groups were chosen to serve as the reference spectrum. After that, the linear regression technique was applied in order to determine the nature of the connection that exists between the eigenvalues and the fundus images. In the end, a transformation matrix was constructed in order to simulate the spectrum of each pixel value present on the image.

PCA is a method that is frequently employed in the field of multivariate statistics. Even though multivariate data undoubtedly provides a wealth of information for a study, if each indicator is analyzed on its own, much of the useful information that the data contains will typically not be fully utilized and will be lost. This will result in incorrect conclusions. As a result, the purpose of PCA is to transform high-dimensional data into low-dimensional data in order to lessen the amount of computation required while maintaining the same level of data accuracy.

Following the completion of the procedures described in the previous section of the experiment, the final step of the experiment was carried out, which consisted of differentiating arteries and veins using the principal components score plots. Following the application of PCA to the data in order to lessen the number of dimensions, a principal component score plot for the first and second principal components of each stage was generated, as is demonstrated in Figure 4. The manually classified images of arteries and veins were contrasted with the unclassified data points on the fundus image and the principal components’ score plot. The veins were shown in blue, and the arteries were shown in red so that the data points could be differentiated. After PCA, the data were normalized so that the distribution could fall between zero and one. This was carried out in order to make the subsequent differentiation of arteries and veins easier. The distribution points of arteries and veins could be distinguished from one another once a threshold was established on the x-axis to serve as the distinguishing condition. After that, the arteries and veins were differentiated based on the conditions for differentiation, and the results were presented on the initial fundus image. It was determined that the portion of the picture that was blue represented a vein, while the portion that was red represented an artery.

## 3. Results

### Average Reflectance Spectrum

In this study, hyperspectral imaging technology was used to analyze the average reflection spectrum of arteries and veins in the retina of patients with DR at different stages, as shown in Figure 5a, based on the degree of lesion (normal, BDR/PPDR, PDR). The results of this analysis were compared to those of patients without DR at the same stages. The average reflectance spectrum of retinal veins can be seen in Figure 5b, which depicts the progression of DR through its various stages. The spectral reflectance intensity of all stages is relatively lower as a result of the difference in thickness between the arteries and veins, which causes the veins to appear darker in fundus images. According to the findings of previous studies, it was observed through other spectral measurement methods that as DR becomes more severe, the venous oxygen saturation will increase in a manner that is proportional to the progression of the condition. From what can be seen in Figure 5b, the difference is most readily apparent during the PDR stage. It is possible to draw the conclusion that the blood vessel blockage causes a reduction in the amount of oxygen that is delivered to the retinal tissue, and that this results in an increase in the blood oxygen concentration of the venous return. As a consequence of this, the overall reflectance trend will gradually increase as the severity of the lesion increases within the red-light frequency band.

The principal components score plots of DR at each stage were used as the basis for distinguishing between arteries and veins, as shown in Figure 6. One of the fundus images of the lesions at each stage was selected to illustrate the results of classifying arteries and veins, as shown in Figure 7. As the severity of the condition worsens, there is a corresponding decrease in the difference in PCA results between the arteries and veins. As a result, it is challenging to determine a threshold that can accurately differentiate between arteries and veins. In addition, when the results of the normal and PDR categories were compared, it was found that there were significant differences between them. These varying results were then analyzed based on three parameters: sensitivity, precision, and F1-scores were utilized in order to evaluate the distinguishment results.

Sensitivity represents the hit rate of the correct judgment, also known as true positive rate (TPR), which is defined in Equation (7).
(7)$$\mathrm{Sensitivity}=\frac{The\;number\;of\;pixels\;judged\;as\;arteries/veins\;correctly}{The\;original\;number\;of\;pixels\;of\;\frac{arteries}{veins}\;in\;the\;fundus\;images}$$

Precision is also known as positive predictive value (PPV), whose values represent the proportions where the locations of arteries and veins are judged correctly. The computation method is shown in Equation (8).
(8)$$\mathrm{Precision}=\frac{The\;number\;of\;pixels\;judged\;as\;arteries\;and\;veins\;correctly}{The\;number\;of\;all\;pixels\;judged\;asarteries\;and\;veins}$$

F1-score is a harmonic mean, and the computation method is shown in Equation (9).
(9)$$F1-score=\frac{2*Sensitivity*Precision}{Sensitivity+Precision}$$

The findings of the investigation are presented in Table 1. The level of DR that was considered normal, BDR, PPDR, and PDR were taken into consideration when selecting the analysis indicator. In the normal, BDR, PPDR, and PDR conditions, the sensitivity values of the arteries are, respectively, 82.4%, 77.5%, 78.1%, and 72.9%. The sensitivities of veins are as follows: 88.5% in the normal range; 85.4% in the BDR range; 81.4% in the PPDR range; and 75.1% in the PDR range.

## 4. Discussion

In this study, the retinal images are classified into four different categories based on a novel HSI algorithm into normal, BDR, PPDR, and PDR. The artery and veins had a precision of more than 80% in the BDR and PDR category, while there was a drop in a precision to 73.4% in PDR in arteries and to 74.6% in veins. There is a general trend toward a gradual decrease in the reflection spectrum as the degree of lesions increases. The development of lesions in the retinal arteries can be attributed to diabetes, which is the cause of this phenomenon [48]. Long-term hyperglycemia in blood vessels causes hemoglobin and platelet aggregation to block blood vessels [49]. Additionally, blood vessels will gradually lose pericytes, basement membranes will gradually thicken, and endothelial cells will be damaged and proliferated as a result of the condition [50]. As a consequence of this, the general trend of spectral reflectance will show a significant decrease in the middle wavelength range (495–570 nm) as the severity of the lesion increases. In addition, it has been discovered that the reflectance of PDR is significantly higher in the higher wavelength (620–780 nm). Because the retinal artery remains in an environment with high blood sugar for an extended period of time, the oxygenated hemoglobin in the blood coagulates, which lowers its oxygen carrying capacity. On the other hand, the non-oxygenated hemoglobin’s capacity to absorb light is improved in the red band. Therefore, as the severity of the disease progresses, there is an increase in both the blood glucose concentration in blood vessels and the proportion of hypoxic hemoglobin. This results in a decrease in the spectral reflectance of retinal arteries in PDR patients within the red band. As the severity of the retinopathy increases, the difference in reflectance between the arteries and veins gradually decreases. This makes it increasingly difficult to differentiate between the two using the PCA results in later stages, as both their precision and sensitivity are reduced. As a direct consequence of this, the precision and sensitivity of HSI algorithm in DR patients who are still in the normal stage or in the PDR stage are, respectively, the highest and lowest levels. In the meantime, the indicator values are comparable between the BDR stage and the PPDR stage because the clinical-pathological severity characteristics of both stages are comparable to one another. In the same stages, the precision of arteries is typically higher than that of veins. This is because arteries contain more hypoxic hemoglobin, which gives them a relatively dark color in fundus images. Veins, on the other hand, contain less of this type of hemoglobin. These factors are easily influenced by the camera, which also results in a poor quality of images captured by the fundus. Additionally, some non-ideal dark areas, which appear in arteries and decrease the value of the hyperspectral algorithm, are misjudged as vein areas. This occurs because arteries tend to have darker areas than ideal. Therefore, the precision of the arteries is superior to that of the veins in each of the four stages of the process. However, due to the relatively higher precision of arteries in comparison to veins, the number of pixels judged as veins in the area of arteries in fundus images is greater than the number of pixels judged as arteries in the area of veins in fundus images. This is because arteries have a relatively higher precision than veins do. Because of this, the sensitivity of arteries is significantly lower compared to that of veins. The number of pictures that were used to train our model is one of the aspects of the study that can be considered a limitation. A total of only 126 images were used in this research. The expansion of the dataset in the future is one of the objectives of this study, which will ultimately result in improvements to the model’s sensitivity and precision.

## 5. Conclusions

In conclusion, the HSI method can be used to distinguish between arteries and veins in retinal images, as well as locate the distribution of arteries and veins within the retina itself. This is possible because of the method’s ability to locate the distribution of arteries and veins within the retina. This function contributes to the establishment of a database for the shifts in blood oxygen concentration that take place across the board in DR’s progression through its various stages. According to the findings, the sensitivity values of arteries are as follows: 82.4% in the normal condition; 77.5% in the BDR condition; 78.1% in the PPDR condition; and 72.9% in the PDR condition. On the other hand, the sensitivity values of veins are as follows: 88.5% in the normal condition; 85.4% in the BDR condition; 81.4% in the PPDR condition; and 75.1% in the PDR condition.

## Figures and Tables

**Figure 1 jpm-13-00939-f001:**
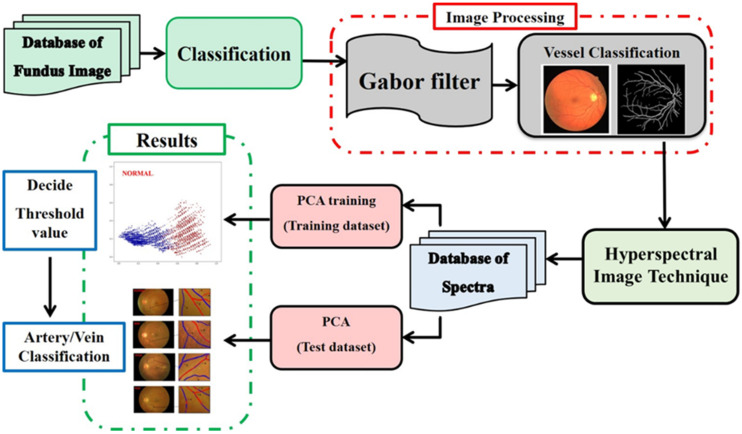
Experimental flow diagram.

**Figure 2 jpm-13-00939-f002:**
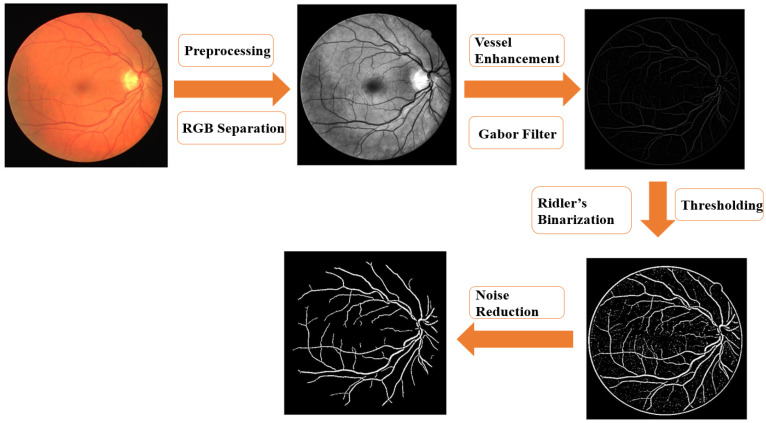
Flow diagram for retinal image processing.

**Figure 3 jpm-13-00939-f003:**
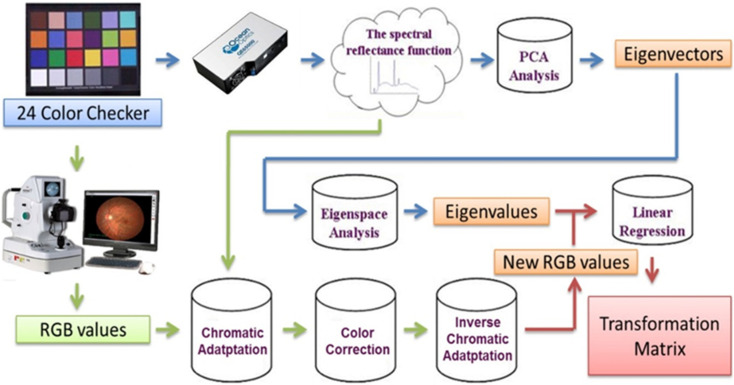
Algorithm flow diagram of the hyperspectral fundus camera.

**Figure 4 jpm-13-00939-f004:**
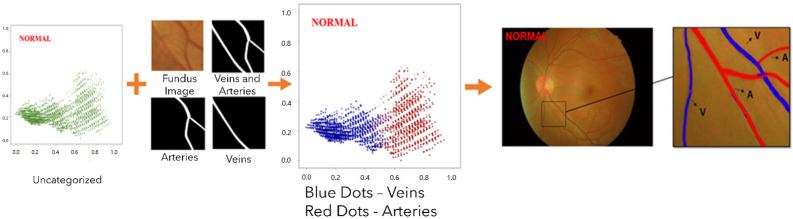
Flow diagram for distinguishing arteries and veins.

**Figure 5 jpm-13-00939-f005:**
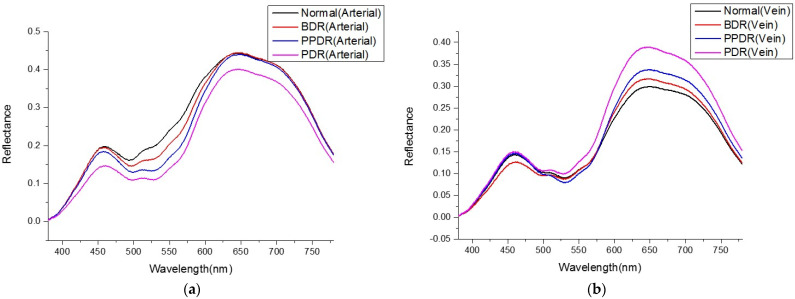
(**a**,**b**) the average reflectance spectrum of normal, BDR, PPDR and PDR DR arteries and veins, respectively.

**Figure 6 jpm-13-00939-f006:**
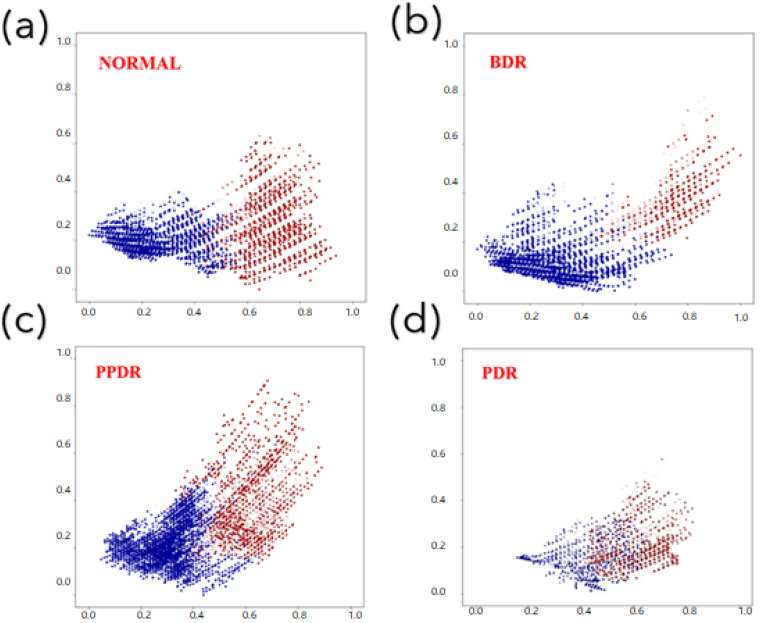
Principle component score plot. (**a**–**d**) show the principal component score of normal, BDR, PPDR, and PDR plots.

**Figure 7 jpm-13-00939-f007:**
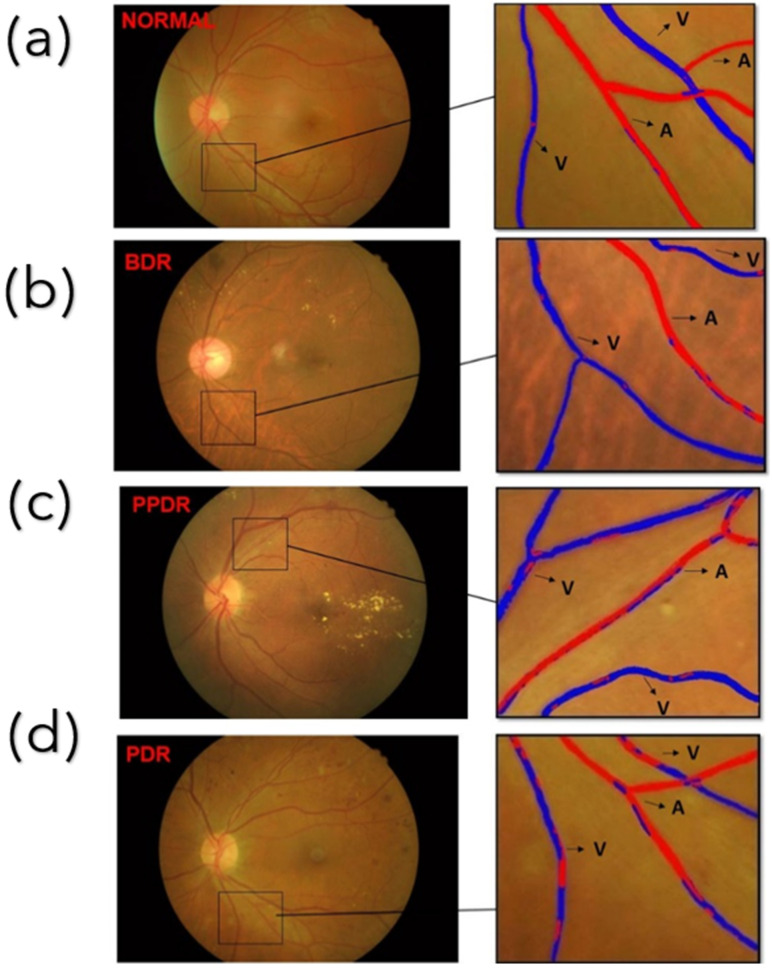
Shows the classification results of blood vessels. (**a**–**d**) show the classification results of blood vessels in the normal, BDR, PPDR, and PDR categories.

**Table 1 jpm-13-00939-t001:** The analysis indicator results of the distinguishment test for arteries and veins.

**Artery**	**Sensitivity (%)**	**Precision (%)**	**F1-Score (%)**
NORMAL	82.4	88.4	85.3
BDR	77.5	83.5	80.3
PPDR	78.1	80.7	79.4
PDR	72.9	73.4	73.1
**Veins**	**Sensitivity (%)**	**Precision (%)**	**F1-Score (%)**
NORMAL	88.5	82.6	85.4
BDR	85.4	81.2	83.2
PPDR	81.4	79.2	80.3
PDR	75.1	74.6	74.8

## Data Availability

The data presented in this study are available in this article.

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
