# Peer review of "Optical Identification of Diabetic Retinopathy Using Hyperspectral Imaging"

_jpm, 2023, doi:10.3390/jpm13060939_

Round 1

Reviewer 1 Report

In this study the idea of the authors is interesting and innovative but the structure of presentation of the work and the poor quality of English especially in some sections such as the abstract make it very difficult to understand.

I suggest to rearrange the work in a more structured way, citing in each section only the necessary.

In the introduction section no data concerning the results should be included and in the results section no obtained datas should be discussed, which instead pertains to the "discussion" section. 

I also recommend not to repeat the data between text and tables: what is inserted in the tables should not be reported in the text and vice versa.

The quality of English in some sections is very poor and makes it difficult to understand the work. I recommend reviewing extensively the quality of the English language especially in section such as abstract. 

Reviewer 2 Report

Dear Authors, thank you for describing such an important topic.

1.line 66:  One of the major reasons for the
loss of vision is DR in the working ages between 20 and 6 [41].

Are you sure "ages between 20 and 6" ?

2.Of course the patient group could be larger (maybe in the future?).

3. I appreciate your photos and  diagrams.

Reviewer 3 Report

This paper should be reviewed by a mathematician and/or imaging scientist.

Being an ophthalmologist specialized in retinal diseases, my concern is if the presented technique has enough sensitivity and specificity in order to be implemented in clinical practice.

lines 75-78: The results should not be presented in the introduction. Please remove them.

In my opinion, the Discussions should include a paragraph stating that at this moment, the sensitivity range (between 75 and 88) do not recommend an application of hyperspectral imaging for identifying retinal blood vessels. What would be the solutions in order to increase sensitivity and precision? Training the algorithm on a larger database of images would help?

The Conclusions should not be another description of the technique, but rather a concise description of the results: sensitivity and precision (and perhaps a phrase regarding future steps that could improve the efficiency of this method)

I do not see the necessity for describing  the principles of fluorescein angiography and OCT in the supplementary file.

There are several problems regarding the English language and also plain errors (typos)

line 63: "as the in its latter stages" - unclear, please rephrase

line 67: "working ages between 20 and 6" - typing error

lines 33, 213, 221: "the precision and sensitivity of diabetic retinopathy patients" - you should not be writing about the precision of the patients, but the precision of HSI in identifying retinal vessels in different stages of diabetic retinopathy!

I strongly recommend to enlist the help of a native English speaker for a complete check of English language and phrasing.

Reviewer 4 Report

The purpose of the study “Optical identification of diabetic retinopathy by hyperspectral imaging” is good, but the description of the article has many shortcomings in general.

Here are some of them, most of them major changes.

Abstract:

Define PDR, BDR and PPDR.

Introduction:

Line 63 remove “the”: However, as in its latter stages, it can cause complete loss of vision.

Line 67: One of the major reasons for the 66 loss of vision is DR in the working ages between 20 and 6 [41]. I think is 60 years, not 6.

Lines 75-78: “The results indicate that the sensitivity values of arteries are 82.4%, 77.5%, 78.1%, and 72.9%, in the normal, BDR, PPDR, and PDR while for veins are 88.5%, 85.4%, 81.4%, and 75.1%, in the normal, BDR, PPDR, and PDR respectively.” These percentages have been taken from this study or are they those stipulated in the ICO? Specify and put bibliographical reference if applicable.

Materials and Methods

What apparatus was used to take images?

What were the inclusion and exclusion criteria to include the images?

Try that table 1 is not cut.

Improve the quality of figure 1 and define PCA int eh figure legend.

Specify the program used for image processing. Have all the steps been done with the same one or have several been used? MatLab does not come out until the conclusions.

Improve the quality of figure 3.

Line 130: Principal components analysis (PCA) is defined in line 99, use the only the abbreviation.

How and where were the numerical data collected to generate the results part? Was a database created?

Was no statistical analysis performed?

On the other hand, it was lacking if the patients were asked for permission to take their images, if there was informed consent and an ethics committee. It should be mentioned in this section.

Results

Lines 154-171 The majority of this paragraph would go into discussion not results. In the results section, only the results obtained in the study itself are explained.

The mean age of each group of patients?

How many men and how many women?

Line 175 "according to past results... " to which results it refers, bibliography is missing.

Figure 5 must be improved.

How long have they been diagnosed with diabetes?

Were all the patients type 1 or type 2 diabetes?

Discussion:

The discussion only comments on the results, it does not compare with the existing bibliography. It is very scarce and seems to be interchanged with what is described in the results that should go in the discussion section.

Conclusion

The conclusion should be brief and concise regarding the results obtained in the study. In this case it is rather another abstract, it is described again until the objective of the study and the methodology carried out.

Supplementary material may serve to augment some part of the work rather than make it so extensive that the reader may or may not look at it. But you should understand the entire article without looking at the supplementary material. And in this article it is in the supplementary material where the device used to take images is mentioned, something important to comment on material and methods.

Round 2

Reviewer 1 Report

After the editing of the paper now it's clearer and well structured.

Author Response

The authors thank the reviewer for such positive comments and in the future the authors would strive to produce quality articles such as this.

Reviewer 4 Report

The purpose of the study “Optical identification of diabetic retinopathy by hyperspectral imaging” is good, but the description of the article has already many shortcomings in general.

Materials and Methods

What company and country is the Zeiss camera from? missing data.

What were the inclusion and exclusion criteria to include the images? Those criteria are not well defined. Just because you have normal vision doesn't mean you don't have another systemic disease.

On the one hand, the inclusion criteria should be set and on the other, the exclusion criteria.

Specify the program used for image processing with details of manufacturer and country.

But, was no statistical analysis performed?

Results

It continues to be an important point since in this pathology the passage of 2 years is crucial. It is necessary to have the exact age of the patients and to know how long they suffered from the disease, as well as to have signed an informed consent that their images will be used.

Still does not put how many men and how many women? that paragraph is not enough.

How long have they been diagnosed with diabetes? This answer is not enough

How is the hospital not going to give data on what type of diabetes it is? they evolve differently and are normally studied separately. This is a serious error.

Discussion:

The discussion is still lacking and does not have bibliographic citations, nor those passed from the results.

Conclusion

The conclusion has improved.

Round 3

Reviewer 4 Report

The article has improved a lot with the mentioned issues corrected.

 Change patients with “normal vision” for control subjects throughout the article (without associated systemic or ophthalmological pathology). Otherwise, it could be a patient with normal vision but who has a serious pathology under treatment that modifies the retina.

 It continues with: "Informed Consent Statement: Written informed consent was waived in this study because of the 378 retrospective anonymous nature of study design." Modify according to what the authors comment that they have the consents.
